# Fatal Unintentional Non-Fire Related Carbon Monoxide Poisoning: Data from Narrative Verdicts in England and Wales, 1998–2019

**DOI:** 10.3390/ijerph19074099

**Published:** 2022-03-30

**Authors:** Rebecca M. Close, Neelam Iqbal, Sarah J. Jones, Andrew Kibble, Robert J. Flanagan, Helen Crabbe, Giovanni S. Leonardi

**Affiliations:** 1Radiation, Chemicals and Environment Directorate, UK Health Security Agency, Didcot OX11 0RQ, UK; neelam.iqbal@phe.gov.uk (N.I.); helen.crabbe@phe.gov.uk (H.C.); giovanni.leonardi@phe.gov.uk (G.S.L.); 2Environmental Public Health Team, Health Protection Division, Public Health Wales, Cardiff CF10 3NW, UK; sarah.jones27@wales.nhs.uk (S.J.J.); andrew.kibble@phe.gov.uk (A.K.); 3Radiation, Chemicals and Environment Directorate (Wales), UK Health Security Agency, Cardiff CF5 2YB, UK; 4Precision Medicine, King’s College Hospital NHS Foundation Trust, London SE5 9RS, UK; robert.flanagan@nhs.net; 5Department of Public Health, Environments and Society, London School of Hygiene and Tropical Medicine, London WC1E 7HT, UK

**Keywords:** carbon monoxide, CO, unintentional poisoning, environmental epidemiology, environmental public health, unintentional death, preventing CO poisoning

## Abstract

Unintentional non-fire related (UNFR) carbon monoxide (CO) poisoning continues to cause fatalities. The narrative verdicts from coroners concerning fatal UNFR CO poisoning in England and Wales, 1998–2019, were collated by the Office for National Statistics. Search terms related to CO exposure were used to obtain information regarding the circumstances of death. Findings were grouped by the location of death, the source of CO, and the reason or behaviour underlying the exposure. There were 750 deaths (77% male). The annual number of deaths decreased over the period studied. Two thirds (68%) of the deaths occurred in the autumn or winter. From the records with information, 59% of deaths occurred within a dwelling (67% male). Males also predominated deaths within vehicles (91%) and garages or outbuildings (95%). From the deaths with information, domestic piped gas was the most common source of CO (36%) and the most frequent underlying factor was inadequate ventilation of exhaust gases (39%, 91% male). Despite the decrease in the annual number of deaths over the study period, there remains a clear need for measures that raise awareness of the dangers of CO poisoning, especially amongst men working alone in garages or outbuildings. Education campaigns and fitting and maintaining CO alarms in high-risk areas should be encouraged.

## 1. Introduction

Unintentional, non-fire related (UNFR) carbon monoxide (CO) poisoning causes hospital admissions and deaths each year in the UK. Such incidents are largely preventable. CO is formed from the incomplete combustion of fuels, such as natural gas or petrol, but may also arise from more insidious sources, such as wood chip being stored as fuel, an apparently cold and used barbeque, and disused coal workings. Once inhaled, CO binds to haemoglobin (to form carboxyhaemoglobin), myoglobin, and mitochondrial cytochrome c oxidase. Not only is oxygen supply to tissues impaired, but oxidative phosphorylation is also inhibited. This sequence of events can lead to potentially permanent neurological damage or death, depending on the severity and duration of exposure [1]. 

According to the UK Cross-Government Group’s 2019–2020 annual report [2] on gas safety and CO awareness, there were 20 deaths per year from UNFR CO poisoning in 2017–2019 in England and Wales compared to the annual figure of 30 deaths per year in 2011–2015 [3]. Similarly, a study using mortality data provided by coroners in England and Wales [4] found that the annual number of such deaths fell from 166 in 1979 to 25 in 2012.

In addition to collating data on fatal UNFR CO poisoning, it is also important to consider the morbidity caused by CO poisoning as this is potentially a burden on health care services too. Roca-Barcelo et al. [5] found an average annual admission rate due to UNFR CO poisoning in England of 250 per year, 2001–2010. A study by the UK National Poisons Information Service (NPIS) based on the analysis of call records, TOXBASE (automated) access, and postal questionnaires reported 1810 UNFR CO exposures from 1 July 2015 to 31 December 2017 [6]. One community study over a 6-month period found that 4.64 CO alarm incidents per 1000 households occurred in private dwellings in one local authority area. Such possible chronic poisoning incidents are difficult to document and are often missed [7]. Finally, UNFR CO data from media reports found that for every fatal incident, another person was often exposed in the same incident but survived, hence prevention strategies could help to reduce morbidity as well as mortality [8]. 

For each death from UNFR CO, several heath care systems are probably accessed. These may include the emergency services, doctors’ certification of death, family bereavement services, and possibly mental health service interventions. In the post COVID-19 situation, where reducing the burden of preventable illness on health and on health services is essential, when a death can be avoided, the need to access all of these things on an already pressurised health care system can also be avoided. 

The clinical and public health effects of CO are well known and covered elsewhere [9,10]. The focus of this study was to establish the circumstances, in terms of the place of exposure, the source of CO, and the behaviour or reason associated with the CO poisoning, as these have not been documented at the population level in the UK. The aim of this study was to analyse narrative data from coroners to gain a better understanding of the circumstances under which fatal UNFR CO poisoning occurred in England and Wales, 1998–2019. 

## 2. Materials and Methods

### 2.1. Data Collection 

UNFR deaths in England and Wales, 1998–2019, that were recorded by the Office for National Statistics (ONS) were studied. Within this, the coroner narrative reports of each incident were gathered. The deaths were classified using International Classification of Diseases (ICD) codes [11].

### 2.2. Data Extraction 

Inclusion criteria for the narrative search on UNFR CO deaths were based on meeting the criteria in Table 1. 

ONS staff were asked to conduct a search of the text of the narrative reports to extract specific items that might provide more information on the circumstances surrounding the death. Such items included demographics (including age and sex), the characteristics of the place of death, CO source(s) in the place of death, and the reason or behaviour underlying the exposure, such as “gas ring left on” or “inadequate ventilation in a garage”. The initial text extracts provided by ONS were studied and three fields for summarising and capturing data were defined. These were: the place of exposure, the source of CO, and the reason or behaviour underlying the exposure. In total, 66 text terms for the place of exposure were allocated into 5 categories for analysis, 137 terms for the source of exposure were also allocated into 5 categorises, and 235 terms for the reason or behaviour underlying the exposure were allocated into 7 categories.

### 2.3. Data Analysis 

The certification of the cause of death can be delayed by up to 18 months when an inquest occurs to establish the cause of death. Therefore, it may be several months before the death is registered. We chose to study the date of death as it is more accurate in terms of season and other time-dependent factors compared to the date of registration. 

To calculate the mortality rate per million population, ONS mid-year population data [12] were applied to each variable, including sex, age group, and geographic region. The rates were calculated for each year and for the total period studied. The median values for each of the characteristics were calculated. 

As the midpoint of this study period, the English index of multiple deprivation (IMD) scores for 2010 were derived from ONS [13] and coded into quintiles using the postcodes of the usual residence for all cases reported from England. For Wales, the 2011 Welsh index of multiple deprivation (WIMD) was used. All cases were coded into the five categories of the national statistics socioeconomic classification (NS-SEC) in STATA using the NS-SEC coding instructions. The codes were based on their standard occupational classifications (SOC) in the coroner records, which utilised the 1990 and 2000 SOC coding as well as additional notes in the free text field of “occupational status” to ensure accuracy. A 3-year rolling average was used to display a smoothed trend in the mortality rate over time. 

Finally, postcode data were used to map the distribution of deaths and derive regional death rates. Postcode data from the usual residence (N = 741) and the place of death (N = 240) were available and both were geocoded, but the latter were excluded from the rates analysis. 

### 2.4. Statistical Methods

Data were analysed using MS Excel and STATA.15. We calculated crude average mortality rates per million population with 95% confidence intervals (CI) by country, region, sex, and age. Age standardised mortality rates for England and Wales were calculated using a direct standardisation method. 

## 3. Results

There were 750 deaths, 77% of which were male (Table 2, Figure 1). Nine deaths had missing postcode data. The annual number of deaths of males and females fluctuated over the period studied, but there was a downward trend with time (Figure 2). 

The median age at death was 57 (IQR 41–74) years. Death rates were highest among those aged 70 and over, specifically among the 80 and over age group, and lowest in the <10 age group. The death rates of males were higher in all age groups except 0–9 and 90+ years, where they were the same as those of females. Deaths were highest in the autumn and winter months (September–February, 68%) compared to spring and summer (March–August, 32%).

### 3.1. Distribution by Country and Region

The highest average annual mortality rate from 1998–2019 was in Wales, with 1.25 deaths per million, followed by the Yorkshire and Humber region in England (0.85). The lowest rate was in southeast England with 0.4 deaths per million. After adjusting for age, the mortality rate for England was unchanged but reduced slightly for Wales (to 1.17 deaths per million).

### 3.2. Deprivation

In England, there was a clear trend of increasing UNFR CO mortalities with increasing deprivation across both sexes. Half (51%) of these deaths occurred in the two most deprived quintiles of the population (four and five). 

### 3.3. Occupational Classification

In England, those in the lowest occupational classification had the highest frequency (32%) of UNFR CO related mortality, followed by the second lowest occupational classification (22%), i.e., 54% of deaths occurred in routine and technical occupations. However, the number of deaths of those in managerial, administrative, and professional occupations was also substantial. The analysis of deprivation and occupational classification among Welsh cases did not provide meaningful data due to the small number of deaths. 

### 3.4. Place of Exposure

There was information on the place of exposure in 74% of the deaths (69% male) and of these, almost 60% of deaths (67% male) occurred inside a dwelling (Table 3). Of the deaths in garages or outbuildings, 95% were male and 91% of the deaths inside vehicles were also male. The percentage of deaths was slightly higher for inside dwellings in Wales than in England, but deaths were proportionately higher in garages or outbuildings in England than in Wales (Table 3).

Around 26% of deaths (91% male) occurred in places where CO alarms were unlikely to be present, such as vehicles, garages, sheds or workshops, containers, portacabins, greenhouses, and tents. The number of deaths in places where CO alarms were unlikely to be present increased to 30% (90% male) when temporary accommodation, such as campervans, caravans, mobile/park homes, and day boats, were included. 

### 3.5. Source of Exposure

There was information on the source of exposure in 63% of the deaths (73% male) (Table 3). Of these, piped gas appliances were the main source of CO (36%, 73% male), followed by petrol or diesel engines (25%, 91% male) and multifuel or solid fuel appliances (such as those that burn coal, wood, smokeless fuels, peat or turf briquettes, and anthracite) (22%, 57% male).

In Wales, the main source of CO was solid or multifuel appliances (44%) rather than piped domestic gas (24%), which was the main source of CO in England (38%). Of the deaths involving domestic gas, 40% involved a central heating boiler, 20% involved a gas cooker, grill or hob, 18% involved a gas fire for space heating, 16% involved a portable gas heater (using a liquefied petroleum gas cylinder as the source), and 7% involved another source. 

In 15 (3%) deaths (93% male), the source of CO was a charcoal barbecue; these deaths occurred 2004–2019. In four (27%) of these deaths, the exposure occurred inside a tent without proper ventilation.

### 3.6. Reason or Behaviour Underlying the Exposure

There was information on the reason or behaviour underlying the exposure in 48% of the records. Of these, inadequate ventilation (39%, 91% male), such as running a vehicle engine inside an enclosed space, was the most frequent cause of poisoning, followed by a faulty appliance being used inside an enclosed space (21%, 65% male). Amongst the records with information for females, the most frequent reason or behaviour underlying the exposure was a faulty appliance (30%), followed by a blockage, such as a blocked flue or blocked air intake (26%). 

In England, the most frequent reason or behaviour underlying the exposure was inadequate ventilation (41%), followed by a faulty appliance (19%). However, in Wales, the most frequent reason or behaviour was a faulty appliance (37%) and then a blockage, such as a blocked flue or blocked air intake (26%). 

## 4. Discussion

We have extracted data from coroners’ narrative reports and described the findings in terms of the place of exposure, the source of CO, and the reason or behaviour underlying the exposure. This was to gain a better understanding of the circumstances under which fatal UNFR CO poisoning occurred, which was the aim of the study and could help in the design of more focused and specific interventions for the prevention of CO poisoning. 

The study found that between 1998 and 2019, there was a steady decline in the annual number of deaths caused by UNFR CO poisoning. A similar trend has been found in other studies in England and Wales [4] as well as in other countries, such as Canada [14] and the United States [15,16]. However, there are still many clearly preventable deaths; for example, those amongst men working alone in outbuildings.

There are many possible reasons for the decline in the annual number of deaths, including increased CO alarm installation. However, it is estimated that only approximately 19% of UK homes have a CO alarm, which is lower than smoke alarm installations [17]. The percentage of UK households with CO alarms installed is much lower than the 29–33% in the United States [18] and around 40% in Canada [14]. Other possible reasons for the decrease include higher living standards, safer appliances, using registered gas appliance fitters, gas appliances being serviced regularly, increased legislation [19], and awareness campaigns, such as the UK CO Awareness Week [20] organised by the Policy Connects communications group. There has also been specific research and reports by the UK All Party Parliamentary Carbon Monoxide Group (APPCOG) [21] and work by CO poisoning prevention charities and Non-Governmental Organisations (NGOs).

### 4.1. Demographics

Death rates were highest amongst males and the over 70s, particularly the 80 and over age group. This is consistent with previous studies [5,18,22,23]. Rates were lower among females, who perhaps undertake less risky behaviour [18]. Older age groups are more likely to spend time indoors, particularly when they are retired, and therefore, are potentially more likely to be exposed inside their dwelling. They are also more likely to suffer from underlying health conditions, which CO poisoning can often exacerbate [17], and may be more likely to be poorer or be living in poorer quality accommodation.

Deaths occurred in all months, with the most occurring in winter (42%). This trend has been observed in many other studies [5,14,18,24,25]. High risk behaviours to keep warm in cooler months, such as using unventilated appliances to heat the home or running car engines inside enclosed spaces, such as garages, are likely factors here. The US has reported an increase in CO related deaths following specific large storms and disaster situations [18]. These increases are often due to an increase in the use of portable petrol or diesel fuelled electricity generators without adequate ventilation [16], using cooking equipment for heating, and a general increase in the use of fuel-burning appliances inside enclosed spaces. However, deaths in the spring and summer months still occur within dwellings and garages or outbuildings. 

Geographically, Wales was the region with the highest average annual rate per million and the difference between that and the rate in England remained after standardising for age. There were also differences between the regions of England, with Yorkshire and Humber reporting the most deaths per million population and the southeast region reporting the lowest. There could be many reasons for these differences and the reported numbers in Wales were relatively small, making it difficult to draw clear comparisons to the data for England.

A study on hospital admissions for UNFR CO poisoning in England found higher rates of admissions in the northeast, with lower rates in the south [5]. Wales was not included in this analysis. They found the east of England and south of England to have the lowest rates: a pattern that was also observed in this study. 

### 4.2. Deprivation and Occupational Classification

The clear trend of increasing UNFR CO mortality with increasing deprivation in England likely reflects poorer housing standards and the poorer maintenance of heating appliances. People living in such accommodation may also be less aware of the dangers posed by CO in domestic situations. 

To reflect the fact that each nation in the UK is responsible for its own health service, efforts were made to analyse the data for England and Wales separately. However, with just 83 deaths in Wales across the study period, deprivation-based analyses were not possible. 

### 4.3. Place of Exposure 

The greatest number of deaths occurred inside the home. Dwellings contain many potential sources of CO and some individuals (such as the elderly) tend to spend most of their time inside their homes. This is consistent with findings from the NPIS [6] and in the US [26].

In recent years, there has been a drive to increase energy efficiency within the home; this has had the knock on effect of reducing ventilation and trapping poisons, such as CO, inside the home [17]. Shrubsole et al. [27] showed that up to 52% of English housing could experience a potential increase in domestic CO concentrations to above the WHO 8-h guideline (10.5 µg·m^−3^) in properties that were subject to energy efficiency measures where purpose provided ventilation was not included as part of the retrofitting scheme.

Education campaigns are essential to also raise awareness of the risk of CO poisoning away from home to ensure that individuals remain alert to potential exposures and consider taking a portable CO alarm with them. The number of men who died in a garage, outbuilding or vehicle points to the need for targeted prevention strategies.

Exposure in temporary accommodation accounted for 9% of deaths. Online accommodation rental company Airbnb, as well as other similar companies, include the presence of a CO alarm as one of their features in accommodation listings [28]. 

### 4.4. Sources of CO Exposure

The most common source of CO poisoning was exposure from domestic piped gas appliances. This was expected as piped gas is the most common fuel used for heating in the UK. It is reported that around 80–85% of the UK population use gas-fired systems (mains gas or liquid petroleum gas from cylinders), although figures do vary by region and country [17,27].

Fewer deaths per million population occurred from the incomplete combustion of gas fuels in Wales compared to England and there were proportionately more deaths involving solid or multifuel appliances. This may be because Wales has many rural areas that are not connected to a piped gas supply. 

Urban areas have higher levels of piped gas supply, with pockets of solid fuel use in Wales and Yorkshire and oil use in Wales, Yorkshire, and the east of England [27]. The Building Research Establishment found in 2007 that overall solid fuel usage was higher in Wales (around 3.5%) than in England (1.4%) [29]. When looking at the combustion of incomplete gas fuels as the greatest source of exposure to CO, gas boilers and heating were the highest contributors. A similar result was obtained by the NPIS survey (faulty boilers, 25.3%) [6]. However, not all poisonings were due to fixed combustion appliances.

### 4.5. Reason or Behaviour Underlying the Exposure 

Of the reports with data, using an appliance or machine with the inadequate ventilation of exhaust gases was by far the most frequent reason for death.

Many accidental CO poisonings and deaths following disaster remediation have been documented [30]. In one study in the US, petrol and disease fuelled electricity generators were the source of CO in 83% of fatal cases [30]. 

### 4.6. Limitations 

Of the data sources that are readily accessible to health professionals, ONS mortality data are relatively more complete and easily accessible in terms of capturing UNFR CO deaths. However, delays of up to 18 months between the date of death and the date of the registration of the death can occur. Thus, it is likely that the UNFR data for CO poisoning deaths that occurred in 2018 and 2019 presented here are incomplete.

It appears there is also uneven reporting by coroners on this topic, with some narrative verdicts being more extensive and providing more detailed information than others. 

Some of the key fields that were analysed had missing data. To obtain the country and region of death, the postcode of the usual residence was used as this information was more complete than the postcode of the place of death. However, the actual location of the death may have been very different from the postcode of the residence. Moreover, when investigating occupational differences, many of the fields for coding SOC90 and SOC20 into NSSEC were incomplete.

From these data, we did not have information on other people involved in the incidents, but work by Fisher et al. [8] demonstrated that for every UK UNFR CO poisoning death, there was on average one other person who almost died. 

There is also a lack of comparability between the English IMD and the Welsh IMD as they include different factors, which made it hard to compare the figures for both countries and is part of the reason why the Welsh IMD was omitted from the results. When analysing the occupational status data, there have naturally been significant changes in the meanings or categorisations of some occupations over the study period in terms of the coding differences between SOC90 and SOC20 and also in terms of the coroners’ recording of certain occupations and the translation of these codes into the NSSEC index, which could not be accounted for in the results. 

### 4.7. Future Research and Public Health Actions

It is important to inform coroners of the results of this work and to highlight its value in improving future data collection. For example, it would be important to record how many of these deaths had a CO alarm fitted in the area where the death occurred in order to inform awareness campaigns. 

Audible CO alarms need to be installed and function correctly to be effective [31]. Moreover, alarms are not the only line of defence. Regular checks and inspections on all gas and vented combustion appliances and fireplaces should be encouraged, as well as consideration regarding the complete household setup. In 2008, a lady died from CO poisoning in a newly built home and her lodger was left in a permanent unresponsive state. Her boiler contained a concealed flue boiler system, which prevented engineers from seeing the whole system, including the exhaust gas flue [32]. As of November 2021, all newly built homes in England require a CO alarm to be fitted when a fixed combustion appliance is installed. By law, all privately and socially rented homes require a CO alarm to be fitted in all rooms where there is a fixed combustion appliance [19].

Within the home, the correct installation, maintenance, and safe use of appliances and the location of CO alarms are important [31]. Vehicles should not be left running inside enclosed areas and unvented combustion sources, such as gas fuelled patio heaters, generators, and barbecues, should not be used inside enclosed spaces [33]. In outbuildings, it is important to raise awareness of the dangers of working alone in confined spaces with inadequate ventilation when a source of CO is present. In temporary accommodation, the dangers of using or storing appliances, such as barbeques (even when apparently cold), and the use of paraffin heaters inside enclosed spaces must be highlighted. Other targeted messages should be delivered in areas with greater deprivation indexes and to those whose first language is not English. It will be important to evaluate the effectiveness of the recent change in the legislation for landlords [19], as well as the legislation for CO alarms on motorboats with living spaces [34]. Intentional CO exposures are the subject of another separate study. 

## 5. Conclusions

Deaths due to UNFR CO poisoning in England and Wales have decreased over time. However, fatalities still occur, especially in certain circumstances, such as in more deprived populations, males working in outbuildings or garages, and amongst the elderly. Prevention strategies should target not only increasing awareness of the dangers of CO, but also the installation and maintenance of CO alarms, especially in garages, outbuildings, and temporary accommodation. The proper installation and regular testing of domestic gas appliances is also important. Coroners have a valuable role in documenting the facts and circumstances associated with UNFR CO poisoning deaths.

## Figures and Tables

**Figure 1 ijerph-19-04099-f001:**
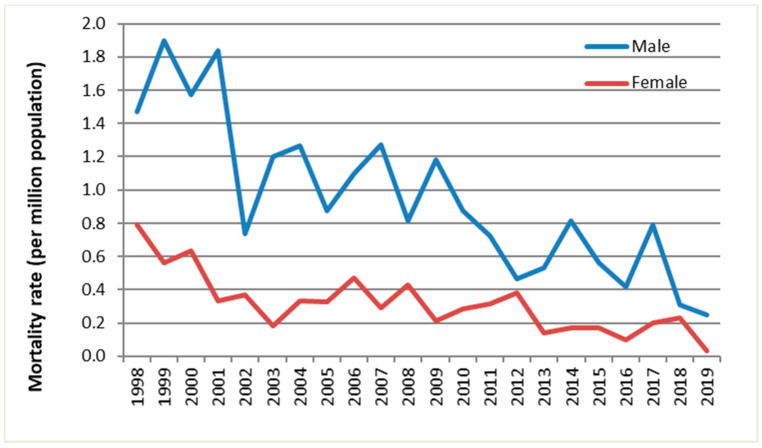
The annual mortality rate of fatal UNFR CO poisoning in England and Wales by sex and year, 1998–2019.

**Figure 2 ijerph-19-04099-f002:**
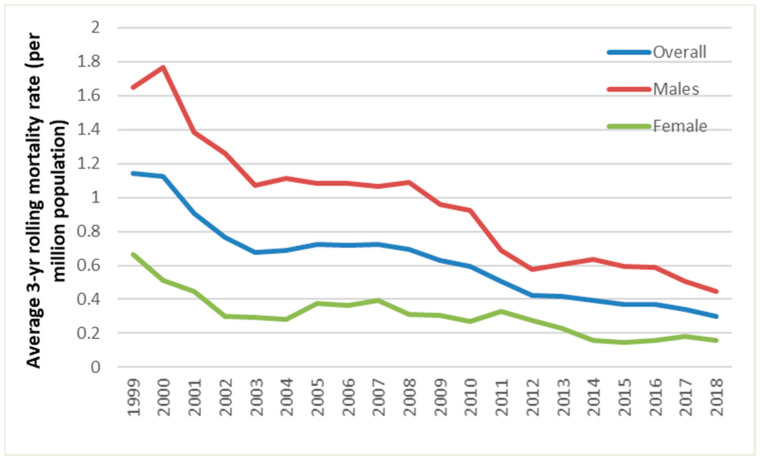
The average 3-year rolling mortality rate of fatal UNFR CO poisoning in England and Wales by sex and year, 1999 to 2018.

**Table 1 ijerph-19-04099-t001:** The International Classification of Diseases (ICD) inclusion criteria for the narrative search on UNFR CO deaths.

Description	ICD 9 Code (1998–2000)	ICD 10 Code (2001 Onwards)
Any death mentioning the toxic effect of CO anywhere on the death certificate	986	T58
*AND the underlying cause was:*		
Accidental ^1^	E800–E869, E880–E929	V01–X59
*OR*		
A disease and no mention of an accidental fire	001–799 and no mention of E890–E899	A00–R99 and no mention of X00–X09

^1^ Excluding accidents that were caused by smoke, fire, and flames.

**Table 2 ijerph-19-04099-t002:** The descriptive characteristics of deaths from UNFR CO poisoning in England and Wales, 1998–2019.

	N (%)	Mean Annual Mortality Rate (Per Million Population)	95% CI	Median Annual Mortality Rate (Per Million Population)
Total	750 (100)	0.62	0.58–0.66	0.61
Male	557 (77)	0.94	0.86–1.02	0.82
Female	193 (23)	0.31	0.27–0.36	0.43
Age Group (years) #				
<10	10 (1)	0.07	0.04–0.13	0.00
10–19	23 (3)	0.16	0.10–0.23	0.15
20–29	51 (8)	0.32	0.24–0.42	0.27
30–39	92 (12)	0.54	0.44–0.66	0.67
40–49	110 (15)	0.65	0.54–0.79	0.25
50–59	124 (16)	0.81	0.68–0.97	1.37
60–69	97 (13)	0.78	0.64–0.96	0.53
70–79	113 (15)	1.26	1.05–1.51	1.29
80+	126 (17)	2.28	1.91–2.71	2.83
Country	741 (100)			
England	658 (89)	0.57	0.53–0.62	0.60
Wales	83 (11)	1.25	0.99–1.55	1.67
English Regions	658 (100)			
East Midlands	46 (7)	0.47	0.34–0.63	0.45
East of England	68 (10)	0.54	0.42–0.68	0.35
London	85 (13)	0.49	0.39–0.60	0.13
Northeast	26 (4)	0.46	0.30–0.68	0.78
Northwest	97 (15)	0.64	0.51–0.78	0.73
Southeast	75 (11)	0.40	0.32–0.50	0.36
Southwest	86 (13)	0.75	0.60–0.92	1.34
West Midlands	78 (12)	0.64	0.51–0.80	0.37
Yorkshire and Humber	97 (15)	0.85	0.69–1.03	0.57

# Age not reported in four deaths.

**Table 3 ijerph-19-04099-t003:** The place, source, and underlying reason or behaviour for exposure of fatal UNFR CO poisoning in England and Wales, 1998–2019 (N = 750).

	N (% of Total)	England N (% of Total) ^$^	Wales N (% of Total) ^$^
**Place of exposure ***			
Dwelling	326 (59)	281 (59)	38 (72)
Vehicle	65 (12)	52 (11)	2 (4)
Temporary accommodation	51 (9)	44 (9)	5 (9)
Garage/outbuilding	111 (20)	99 (21)	8 (15)
**Source of CO ****			
Domestic piped gas supply	170 (36)	150 (38)	13 (24)
Petrol/diesel	117 (25)	104 (26)	6 (11)
Solid/multifuel ^^^	102 (22)	77 (19)	24 (44)
Other	83 (17)	69 (17)	11 (21)
**Reason or behaviour underlying the exposure *****			
Inadequate ventilation	153 (39)	138 (41)	6 (13)
Faulty appliance	82 (21)	64 (19)	17 (37)
Blocked flue or air intake	65 (17)	53 (16)	12 (26)
Poor maintenance of appliance	45 (12)	36 (11)	9 (20)
Behaviour—no malfunction identified ^&^	32 (8)	30 (9)	2 (4)
Leakage from an appliance or exhaust system	13 (3)	12 (4)	0

* 197 (26%) no information; ** 278 (37%) no information; *** 360 (48%) no information; $, 32 missing postcodes; ^ Solid/multifuel includes the burning of coal, wood, smokeless fuels, peat or turf briquettes, and anthracite; &, For example, using a gas grill with the door shut or a gas cooker left on.

## Data Availability

Link to the ONS mid-year population estimates data: https://www.ons.gov.uk/peoplepopulationandcommunity/populationandmigration/populationestimates (accessed on 10 January 2022).

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
