# Peer review of "Fatal Unintentional Non-Fire Related Carbon Monoxide Poisoning: Data from Narrative Verdicts in England and Wales, 1998–2019"

_ijerph, 2022, doi:10.3390/ijerph19074099_

Round 1
Reviewer 1 Report
This is a retrospective review of coroners’ reports of fatal CO poisoning in England and Wales to attempt to determine circumstances and risk factors that might be modified to prevent future poisoning. The paper is well written and provided in publication format.
Line 31, 225 – Suggest changing “outhouses” to “out buildings”
Lines 39-41 – This paper is about fatalities. How does modifying their number reduce the burden on the health services?
Line 47 – What about other toxic effects of CO other than COHb formation?
Lines 48-52 – Why are these two estimates so different?
Results presentation is appropriate.
Lines 226-235 - Could increasing use of automotive catalytic convertors be playing a role?
Much discussion is given to the installation and use of CO alarms, apparently required in new construction with fixed combustion appliances. Is it possible to estimate how many deaths would have been prevented had they been required in all residences? Not all poisonings were due to fixed combustion appliances.
Reviewer 2 Report
Review for article “Fatal unintentional non-fire related carbon monoxide poisoning: Data from Narrative Verdicts, England and Wales, 1998- 2019”
The idea of the subject is attractive; however, the scientific value of the study is not so high. Actually as an Original Research article, it cannot receive a good acceptance. While as a Review or Communication Paper it can receive acceptance after following revisions made:
- Part Introduction needs extension. It seems this part is uncompleted not providing proper history and literature for such a huge subject.
- The box in lines from 77 to 93 cannot be categorized in a proper article format. It should be written as a table, a supplementary file or an Appendix after Reference part.
- As dear authors know, CO can have various health effects according to its concentration. Death is the highest effect and actually the most important one. Please prepare a table about all CO health effects in different concentrations and note the predominant pollution sources and the most vulnerable risk receptors for each concentration generally. After this part, if authors provide specific data and real statistics for such health impacts as extension of their case study, it can advance the article impact as a comprehensive study.
- Please add some notes about managerial, engineering, governmental and health care resolutions for overcoming the problems engaged with UNFRCO. What are the best methods for detection of CO concentrations in different places? What are the best ventilation systems for such concentrations? Describe in details. Please prepare a prominent roadmap for guiding the highlighted resolutions in an attractive format for enhancing the article impact.
- Discussion and Conclusion parts need more words to determine the main reasons, necessity and goals of the study.
Reviewer 3 Report
I had the privilege to review this paper detailing features of CO related deaths in Enlgand and Wales.
I think the paper is well-written and sufficiently descriptive regarding the available information arising from the datasets consulted.
I'd only be interested in whether the authors tracked suicidal intents in their search, and if not why. This should be highlighted since suicide is another important cause of death involving CO (garages, vehicles..).
Again, good job. I hope further work and data will be collected prospectively by involving GO and NGOs/charities and opening a specific register.
